# A Qualitative Assessment of the Essential Health and Nutrition Service Delivery in the Context of COVID-19 in Bangladesh: The Perspective of Divisional Directors

**DOI:** 10.3390/healthcare10091619

**Published:** 2022-08-25

**Authors:** Pablo Gaitán-Rossi, Mireya Vilar-Compte, Valeria Cruz-Villalba, Nazme Sabina, Manuela Villar-Uribe

**Affiliations:** 1Instituto de Investigaciones para el Desarrollo con Equidad, EQUIDE, Universidad Iberoamericana, Ciudad de México 01219, Mexico; 2Department of Public Health, Montclair State University, Montclair, NJ 07043, USA; 3DevResonance Ltd., Dhaka 1212, Bangladesh; 4Health, Nutrition and Population Global Practice of the World Bank Group, Washington, DC 20433, USA

**Keywords:** Bangladesh, COVID-19, health service provision, essential health and nutrition services, Primary Health Care System Framework, primary health care

## Abstract

Bangladesh suffered disruptions in the utilization of essential health and nutrition services (EHNS) during the COVID-19 pandemic. The magnitude of the pandemic has been documented, but little is known from the perspectives of health administrators. A rapid qualitative assessment of division-level capacity identified successes and bottlenecks in providing EHNS- and COVID-19-related services during the first months of the pandemic in Bangladesh. Semi-structured interviews were held with the Health and Family Planning Divisional Directors of the Ministry of Health and Family Welfare. The Primary Health Care System Framework guided the content analysis, focusing on (i) service delivery, (ii) communication and community outreach, and (iii) surveillance and service monitoring. Our findings identified low care seeking due to fears of getting infected and unawareness that EHNS were still available. Adaptations to telemedicine were highly heterogeneous between divisions, but collaboration with NGOs were fruitful in reinstating outreach activities. Guidelines were centered on COVID-19 information and less so on EHNS. The inflexibility of spending capacities at divisional and clinic levels hindered service provision. Misinformation and information voids were difficult to handle all around the country. Community health workers were useful for outreach communication. EHNS must be guaranteed during sanitary emergencies, and Bangladesh presented with both significant efforts and areas of opportunity for improvement.

## 1. Introduction

During the last few decades, Bangladesh has made remarkable achievements in its population health, including improvements in life expectancy and under-five mortality rates [1,2]. Bangladesh’s political commitment to expanding access to essential health and nutrition services (EHNS) has played a key role in achieving such positive outcomes [3]. However, a recent study [3] revealed that important challenges remain, including: the quality of care and equitable access to services, the availability of essential inputs and health workers, issues around facility organization and management, and a fragile coordination in the provision of primary health care services, resulting from the fragmentation in the health system and an inflexible health financing system.

Given the existence of such challenges, when the COVID-19 pandemic struck the country, there was an immediate concern of international agencies that this could imply an enormous strain on Bangladesh’s health system. 

As in the case of other countries, Bangladesh declared a national lockdown from 26 March to 30 May 2020 [4]. This measure was complemented by a preparedness and response plan issued in July 2020 by the Ministry of Health and Family Welfare. At the national level, instructions and guidelines for addressing essential health and family planning services during the pandemic were developed [5] and distributed—from health divisions, then to district hospitals, and finally to local primary clinics (upazilas, unions, and wards)—to homogenize service delivery and ensure the safety and capacity of the health services during the pandemic [6].

Resulting from the aforementioned changes in health care provision and the immense pressure brought by the pandemic response, as well as the pre-pandemic challenges faced by the country’s health system, in 20220 Bangladesh experienced problems in EHNS on both the supply and demand sides. Between April and May 2020, when the pandemic cases started to climb [5], administrative data indicated sharp reductions in services across all levels of care [7], including primary, secondary, and tertiary care. There were problems in the provision of services related to the disruption of supply chains of medicines and contraceptives, reduced hours for EHNS at clinics and hospitals, strains on human resources for health, and the reallocation of resources to COVID-19 related activities [8,9]. Prior evidence suggests that this severely affected antenatal and postnatal care (ANC and PNC), which dropped at the onset of the lockdowns in most of the country [4]. Qualitative studies in areas such as Khulna and Dhaka [10,11] suggest that, during the early stages of the pandemic, there were personal and organizational barriers that contributed to the disruption in the provision of health care services, including higher workloads and fatigue, psychological distress among health providers, a shortage of good-quality personal protective equipment (PPE) that increased fears of infection, and on-site management problems.

From a demand perspective, the existing literature [12] suggests that Bangladesh suffered from challenges similar to those observed in other countries, such as the avoidance of health facilities due to fear, lockdown effects on mobility, unawareness of service availability, and a reduced ability to pay for health services [13]). In fact, the BRAC survey [12] reports that two in five households refrained from seeking healthcare due to fears of infection, and among specific subgroups, such as pregnant women, findings reveal the enormous impact of the pandemic during the initial months, with individuals not using regular healthcare services (11%), receiving less than four ANC visits (38%), delivering at home (29%), and not attending PNC (39%). These disruptions have led to worrisome impacts, including a decrease in vaccination (i.e., one-fourth of newborn babies were not vaccinated during the pandemic [12]) and increased child deaths (12.8%) and maternal deaths (7.6%) [9]. Similar disruptions have been reported in other studies focusing specifically on rural sub-districts or urban areas [14]. In addition, a qualitative study of 15 pregnant women in Khulna revealed how they experienced socioeconomic hardships (loss of income, difficulty in receiving loans, and a lack of transportation) and emotional distress (fear, worry, and sadness [10]).

The available evidence shows a remarkable effort to document EHNS disruptions in Bangladesh from the supply and demand perspectives. There is information on the government’s plans, administrative data on service provision and utilization, population surveys on service barriers, and interviews on the experiences of healthcare providers and EHNS users, especially those associated with maternal and child services. However, there is scant evidence of the experiences of health administrators at the divisional level during crucial months of the pandemic. Their perspective matters, because they operate using the national standardized guidelines between the local hospitals and clinics where they need to be implemented. Moreover, their accounts fill information gaps on service disruptions and adaptations that have not been explored as thoroughly as maternal and child services. Hence, with the available evidence, it is important to document and triangulate their experiences in providing EHNS- and COVID-19-related services using the narratives of those in charge of managing these services.

## 2. Materials and Methods

In the early months of the pandemic, the World Bank conducted a rapid qualitative descriptive study to document the experiences of division-level directors during the first months of the pandemic in Bangladesh. This was a rapid response based on the fears of the enormous strains that the pandemic could bring to the health system, which were anticipated based on the challenges that a recent study had highlighted [3]. The goal of the study was to provide information to the government and key stakeholders rapidly using the Primary Health Care (PHC) Resilience Framework [15]. Semi-structured interviews with the Divisional Directors of Health and Family Planning were conducted during the initial months of the pandemic, and the material was then transcribed and coded for interpretation. The results were discussed in a workshop organized by the World Health Organization regional office.

The qualitative description method consisted of a ‘naturalistic inquiry that uses low inference interpretation to present results in everyday language’ [16]. In this research context, this descriptive methodology aimed to facilitate the understanding of health outcomes and their interactions with the health care system and sought to improve clinical care and health systems [17]. Traditional qualitative research usually requires long periods of time for the data collection and analyses. In contrast, rapid qualitative techniques have been developed to work in environments requiring the fast sharing of findings. For instance, these techniques are used in research that informs needs, priorities, and adjustments in policy and practice in contexts where health landscapes and health organizations change quickly during the study period [18]. Some of the features of these techniques are: (a) data collection in a short period of time, conducted iteratively as the context changes; (b) a focus on the insider´s or the *emic* perspective, which tends to be participatory and involves feedback; (c) the triangulation of data and available evidence; and (d) a reliance on a large team, with each division performing simultaneous specific tasks [19]. These techniques were recommended and used during the COVID-19 pandemic to accompany traditional epidemiological approaches to help us to understand not just the “what” but also the “how” in terms of the ways that health services are being affected. One of the main contributions of this research is its focus on medical response experiences during outbreaks and disasters [20].

The research design, the interview guides, and the content analysis of the study [21] were centered on the PHC System Resilience Framework for the provision of EHNS. The PHC System Resilience Framework presented by the World Bank [15] is an analytical framework that posits that PHC can contribute to pandemic preparedness, responses, and recovery while ensuring EHNS delivery by maintaining and reinforcing three interconnected core functions:(1)*Service delivery* is the ability of the PHC system to concurrently deliver care related to the emergency and routine essential health services. Emergency-related PHC services, during the COVID-19 pandemic, included the basic treatment and follow-up care for patients with mild symptoms, the provision or facilitation of laboratory tests, triage, referral to hospitals, and mental health services. Routine essential health services typically include services for mothers and children (such as family planning, child vaccinations and growth monitoring, sick childcare, and antenatal, maternity, and postnatal care), infectious disease services (such as services for HIV, TB, and sexually transmitted infections), and noncommunicable disease services (for example, continuous services for diabetes, hypertension, and respiratory and mental health care).(2)*Communications and community outreach* refer to the PHC system’s capacity to establish and maintain a dialogue with the community to generate trust, promote healthy behaviors, provide actions for the prevention and control of emergencies (including the provision of accurate information on disease spread, vulnerable populations, and preventive measures such as handwashing and physical distancing), and promote the continued use of essential services for care continuity (the promotion of vaccinations for children, antenatal care, family planning services, and diabetes screening and management, among many others, coupled with information on health care access and the assurance of health care safety). Communications also refer to the ability of the PHC system, its health care workers, and its facilities to communicate with other actors in the health system (including public health institutions and hospitals) in order to ensure coordination throughout the care pathway, as well as with other sectors involved in the provision of emergency-related or social services, such as transportation, education, and social protection.(3)*Surveillance and service monitoring* mainly relate to the continuous collection and reporting of high-quality data on the emergency-related overall disease burden and on the services delivered to the population. The continuous monitoring of the disease burden and service delivery help national and local-level decision makers to better adjust their care models and necessary inputs, such as extending service hours, increasing necessary human resources, medications, or personal protective equipment, and adjusting outreach service models. For emergencies due to infectious disease outbreaks, PHC systems can play an essential role in data collection and reporting activities (passive surveillance), and in testing, contact tracing, and isolation management activities (active surveillance) for epidemic control.

The present study focuses on the *emic* perspectives of all the Health and Family Planning Divisional Directors of the Ministry of Health and Family Welfare (MOHFW) from each of the eight divisions of Bangladesh. Sixteen key informants completed interviews, including eight divisional directors from the Directorate of Health and eight divisional directors from the Family Planning Directorate. Divisional directors offer a unique insider´s perspective that can be used to assess the PHC System Resilience Framework, because they provide administrative guidance to the district-level health and family-planning authorities, and they are responsible for the supervision, monitoring, and coordination of health activities. During the COVID-19 pandemic response, divisional directors implemented instructions from the national committees and facilitated logistic and financial support to raise awareness and quarantine the population when necessary [22,23]. Thus, fully understanding the pandemic response in Bangladesh requires an account of how they experienced a key period of EHNS service disruptions.

The interviews were based on a semi-structured interview guide, which can be found in Appendix B, and were conducted in Bangla by a local researcher. Interview questions were collaboratively developed, following the PHC Resilience Framework, between the local consultants and the World Bank, with the overarching aim to understand the events, perceptions, contexts, and narratives of key informants. Interviews were conducted virtually via video call or telephone interview between 22 February 2021 and 16 March 2021, with an average call duration of 42 min, as is common with rapid techniques that avoid placing a research burden on participants currently working on the pandemic response [19]. In most cases, the interviews were recorded and subsequently transcribed and translated into English for analysis. Since we were able to interview all our key informants, we consider that we reached saturation on most of the themes we aimed to address—information gaps are highlighted in the results section. The study only collected data from key informants; therefore, none of the involved institutions required Institutional Review Board approval. Moreover, the study was approved by the World Bank and by health authorities in Bangladesh [24] (report no: PIDC29865). Nonetheless, the study adhered to ethical standards, such as voluntary participation, an oral informed consent preceded by an explanation of the study objectives, guarantees of anonymity, secure data storage, and assurances that the benefits would outweigh the possible damages of the study. We did not offer any incentives to participate.

When analyzing the interviews, based on the PHC Resilience Framework, codes and sub-categories were developed to identify the facilitators and barriers associated with the provision of EHNS and to organize and narrate the divisional directors’ experiences. Two research team members independently coded the 16 interviews, while a third researcher supported any questions through discussion. The coding was performed using a line-by-line approach. All analyses were conducted in Dedoose Version 8.3.47, a web application for managing, analyzing, and presenting qualitative research data.

All results reflect the perspectives of the interviewees and received positive reactions during the workshop with international organizations that have been heavily involved in the pandemic response in Bangladesh and that work hand-in-hand with ad hoc governmental agencies, indicating that the processes described mostly matched their own accounts. The Appendix A section shows the key findings that guided the processes described in the Results section in greater detail and offers additional examples of significant quotes.

## 3. Results

Key informants were primarily male (87.5%) and had a mean age of 55.8 years (SD = 1.8). Most of them (43.75%) had held their position for 1–3 years and 50% of them were medical doctors. Full demographic information on the key informants is provided in Table 1.

### 3.1. Disruptions in Service Delivery and Community Outreach

During the initial months of the pandemic, most services were disrupted across all divisions, and the provision of services mainly focused on emergency and COVID-19-specific care. In fact, according to the interviewees, the government officially discouraged non-emergency visits to clinics and fostered remote ways of getting in touch with patients. This strategy led to several interruptions in service delivery, including vitamin supplementation, in-person examination during pregnancy, tuberculosis screening and treatment, and services for chronic conditions.

Community outreach service provision and activities were also reduced. Emergency and essential services, such as home visits to provide contraceptives and vaccination campaigns, continued; however, scheduled outreach events, door-to-door visits, and courtyard meetings were cancelled or postponed due to social distancing measures. For example, an informant mentioned that most of their community outreach services are traditional, informal, and rely on physical contact. Therefore, those services shrank in order to comply with the social distancing measures. Community health workers (CHW) played a vital role in outreach by communicating COVID-19 information, visiting households from the target population, motivating people to access services, and organizing catch-up campaigns. They benefited from being trusted sources of information in the communities. However, CHWs reported fears at the beginning of the pandemic, mainly due to deaths amongst them. These fears subsided after a few months (June), by which time they understood safety protocols. Partnerships with NGOs and religious institutions were key. NGOs supported the strained health system by coordinating awareness programs and communicating key messages about, for instance, preventing unwanted pregnancies. In addition, several divisions joined forces with imams—in a general sense, persons leading Muslim worshippers in prayer—to raise COVID-19 awareness, communicate safety protocols, and encourage service utilization.

### 3.2. Contextual Factors Hampering Service Delivery

Confinement measures implemented during the general lockdown period (26 March–31 May 2020) had important consequences for service delivery by limiting services providing transportation to clinics for patients, and for the distribution of medical supplies, since movement restrictions and travel bans were imposed. Special transportation arrangements were needed to ensure that supplies were delivered to different areas. Furthermore, an informant stated that the strict transport restrictions during April and May and the inability to leave the containment zone prevented pregnant women from going to their routine visits. In addition, natural disasters, such as floods (during August and September 2020), hampered the provision of services in Rangpur and Sylhet.

### 3.3. Low Care-Seeking Due to Fears of Getting Infected with COVID-19

There was a general perception that people would be safer at home, even if services such as delivery or vaccines were needed, and that attendance to a clinic would be risky. As one health director stated, “at the beginning (from the end of March to April), people did not visit health care centers, fearing the COVID”. In fact, families that needed to hospitalize a member often requested early discharge. There was a belief that health workers were carriers of the virus, which led to lower care-seeking due to fears of getting infected. In Khulna, where 200 doctors and field-level healthcare assistants became infected with COVID-19, such fears caused pregnant women to deliver their babies at home, and in Sylhet, families discontinued newborns’ care. In fact, some health providers suffered stigmatization. These beliefs and sentiments of fear might have been reinforced by (social) media or mouth-to-mouth communication propagating the perception that COVID-19 was spreading in clinics through infected personnel.

### 3.4. Misinformation

Misinformation, which mainly spread through social media, was a problem both for health providers and for the general population in most divisions. Misinformation stemmed from communication voids and exacerbated the notion that hospitals and clinics were only for COVID-19, leading to a sharp decrease in the utilization of services. Authorities were unable to prevent the misinformation and could not manage it properly: “it was beyond our capacity to control social media and the internet”. They did not have any mechanism for addressing misinformation panic, either. Health providers and other actors tried to reach out to patients and communities to explain that health services were still available and that they should continue to seek care despite the COVID-19 pandemic. CHWs played a crucial role because they knew the members of the communities. However, informants underscored that there was generalized panic in the population that affected perceptions and the utilization of services. In two divisions, social media or awareness campaigns were put in place to counteract some of these barriers; however, it is currently unclear whether these were effective. Eventually, an aspect that helped to mitigate the fear of COVID-19 was the low mortality rates that were observed.

### 3.5. Technological Solutions for Telemedicine

The government encouraged the use of technology for telemedicine to support service provision. Technology was used in different forms and exhibited previously observed positive trends. Phone calls, short message services (SMS) through mobile phones, and social media were key tools used to communicate with the population. As highlighted by an informant, “most people have access to mobile phones”, which can be used to contact doctors. Facebook was also used as a resource to enable patients and providers to communicate. Doctors posted their phone numbers online on social media platforms to facilitate people’s ability to contact them for service provision, which was effective in reaching out to those that needed attention. Many providers also shifted to online services through Zoom. However, this shift was not always easily accomplished, as budgets and connectivity were not always available. Online services were often paid out-of-pocket by providers directly. Moreover, several divisions complained of the scarce guidance on how to provide telemedicine services. In addition, Zoom was used to coordinate service-related activities between health professionals.

Some divisions reported using information systems—which are commonly used to record process outcomes related to services, staff, and locations—to monitor logistics and service provision. However, two informants mentioned that they did not have access to recent health data about their divisions and complained that there was “no formal and organized survey data for our Division or Directorate”. Likewise, they underscored how the office did not have “any mechanism to collect information”.

Flyers and leaflets were also common means of communication. Outreach campaigns used vans to communicate messages about unwanted pregnancy prevention during COVID-19 and family planning services. However, only a few divisions reported using this strategy, and, in some places, megaphones were not authorized.

### 3.6. Availability of Human Resources

There were significant gaps in the availability of specific cadres of human resources. Several informants expressed the need for specific human resources, such as public health experts and virologists, amongst others. This is an ongoing problem that was magnified by the pandemic, which requires the performance of new tasks in addition to routine ones. As highlighted by an informant, “workload has increased as never before”, affecting clinical and community services. In the same vein, one informant relayed the following account when describing health staff: “they were superhuman beings as they had to perform all of their own and others’ duties as well”. While health professionals worked with determination, provider exhaustion increased as they assumed the burden of extra tasks. Health professionals faced unexpected challenges that disrupted their performance in health provision and placed pressure on their workload, such as colleagues getting infected or even dying. As observed in other countries, such extra efforts have imposed a heavy psychological burden and an increased risk of infection and have compromised work environments, leading to provider strikes demanding better pay. While the government promised an incentive, this had not been fulfilled at the time that the interviews for this study took place.

### 3.7. Training and Supervision

In terms of training, there were two important phases linked to service delivery. In the initial months of the pandemic, training focused on aspects such as using protective equipment, testing, etc. As services resumed in the later stages of the pandemic, training was targeted at ensuring safety measures in service delivery. Guidelines played a role in the continuity of EHNS delivery across divisions, including public health interventions. However, the narratives of directors were often unclear, as they suggested that they received various guidelines, but they rarely mentioned specific ones by name. Guidance was also centered on COVID-19 information and safety protocols among health providers. While such guidelines were useful, informants also emphasized that some of them backfired and complicated service provision. For example, the request for COVID-19 tests before going into labor when testing availability was low reduced incentives to deliver in health facilities.

Outreach messages and guidelines for communicating with the population were defined by national authorities and mostly contained COVID-19 information, but they were not tailored to specific needs. While some NGOs directed messages towards specific populations to motivate people to use ANC and PNC services, as well as the delivery of institutional services, there were areas, such as maternal health, for which tailored guidance was not provided. This lack of outreach messages and guidelines led to controversial recommendations, such as a recurrent request to avoid pregnancies during the COVID-19 pandemic.

Regular in-person supervision decreased during the first months: “by half during the year; none during lockdown”. Prior to the pandemic, supervision relied on face-to-face meetings and checklists, but during the pandemic they had to substitute these strategies for virtual meetings, relying heavily on virtual platforms, such as Zoom, phone, and text messages to do so.

### 3.8. Financing Restrictions

In the initial stages of the pandemic, several divisions stated that protective equipment arrived late and that it was of low quality. Given the risk of contagion, providers often decided to buy their own protective equipment through personal resources, as divisions and clinics did not have flexibility in spending decisions. Slowly, distribution measures and budgets adapted to the new needs of COVID-19, including aspects such as sanitizers, gloves, etc. This adaptation was reinforced by donations of protective equipment from NGOs and individual donors, and the improvement of the quality of such equipment.

The pandemic also highlighted the inflexibility of budgets for human resource adaptations according to local and contextual needs. Limitations for personnel recruitment worsened workloads and the exhaustion of health professionals. There were limitations in human resource management, specifically related to the lack of autonomy of divisions in recruiting personnel as needed during the emergency and filling vacant positions. More specifically, the inflexibility of budgets meant that the adaptations to human resources required by the pandemic were unfeasible, leading to solutions such as the hiring of paid volunteers.

Two divisions stressed that they had insufficient infrastructure to deal with the pandemic and no budgetary alternatives to face it. Divisions faced challenges in coping with paying for the new technologies needed for service provision and coordination. In some divisions, the costs of new technologies were addressed by the office, while in other divisions, payment for new technologies was covered by providers’ out-of-pocket financial contributions. Some CHWs reported the need to pay for unexpected costs (i.e., the internet access required for telemedicine).

### 3.9. Return to a More Stable Service Provision

Even as lockdown measures have relaxed, the provision of services has not been easy to implement at clinics. Returning to a more open service delivery mode has required clinics to follow COVID-19 protocols that are difficult to enact, including the need to cope with EHNS- and COVID-19-related services in parallel. This has been particularly challenging in health facilities that have inadequate infrastructure to implement protocols (i.e., isolation areas) without disruptions to spaces and human resources for routine care. A positive aspect has been the sustained collaboration with NGOs in order to reinstate services, including outreach activities.

After the initial lockdown phase, community outreach services resumed, albeit with service modifications to adapt to the COVID-19 safety protocols. Door-to-door visits and outreach events resumed when the danger was perceived as ‘low’ based on the low death rates. Outreach sessions originally involved social gatherings but were modified to comply with COVID-19 guidelines for social distancing.

### 3.10. Summary of Barriers

The narratives of the directors helped to identify important barriers to service provision and illustrate which of these were common among divisions (Table 2). On the demand side, the most frequent barriers were the fear of going to the clinic or a hospital due to the risk of getting infected and transportation restrictions. From the informants’ perspective, this fear was likely fueled by widespread misinformation. While these two factors were common throughout all divisions, contextual factors affecting transportation were shaped by local characteristics (i.e., floods).

The prominent barriers on the supply side were the service delivery disruptions, shortages of health providers, and the late arrival and low quality of protective equipment. In addition, guidelines with insufficient information and clarification formed a common narrative among directors from most divisions.

The rest of the barriers were not salient in most divisions and thus responded to local dynamics. They could be present elsewhere, but the directors did not place a particular emphasis on them. Nevertheless, they reveal the heterogeneity of the needs and responses. For instance, the need to tailor the guidelines to specific populations was more important in some divisions than in others. Likewise, the use of technology such as information systems was common in some divisions, but others complained that they did not have updated tools and information.

## 4. Discussion

The results suggest that the continuity of EHNS could not be ensured in Bangladesh during the pandemic, especially in the initial months. This correlates with findings from administrative data [7] and longitudinal surveys [25], as well as findings reported in other countries [26,27]. Demand and supply limitations for service delivery during lockdown, difficulties in communication, and limitations in the local monitoring of the disease and service delivery were some of the primary causes of service disruptions. While some of these problems have already been noted in the region [13], the divisional directors’ accounts offer a more nuanced perspective on how they experienced these problems inside health care services and thus offer specific examples for improving the pandemic response. The pandemic led to different and changing needs in terms of human resources, training, and technology that the administration processes could not address in a timely and adequate manner. Therefore, at least during emergencies [28], such processes require a better response, as shown by recommendations to build resilient health systems [29].

An important finding is that this analysis did not reveal distinct patterns as per the division. In fact, there were relevant problems which were faced nation-wide, with only small differences identified at the division level. Likewise, epidemiological data at the district level confirm major drops in ANC, PNC, and other types of deliveries between February and April 2020 across the whole country, and a moderate recovery by July 2020 [4]. Despite differences across divisions in terms of COVID-19 and health care services capacities, such homogeneity might have resulted from the implementation of top-down measures at the national level, in which divisional directors played an important role.

Common supply and demand limitations were identified. Supply-side limitations included disruptions in service delivery, especially during the initial months of the pandemic, as services mainly focused on tackling the COVID-19 emergency. Outreach services and activities were also disrupted, as was reported by the BRAC population survey [12]. However, what the directors reported were the internal struggles that explain these disruptions. The lockdown measures affected service supply through indirect mechanisms, such as limitations in transportation, which negatively impacted patients’ ability to reach services, as well as the distribution of medical inputs. The directors’ narratives also extend beyond the lockdown. For instance, they explained that, once the lockdown measures were relaxed, health services still faced supply limitations due to the protocols that had to be enforced for safety reasons and oftentimes required unfeasible human resources and physical infrastructure to be executed. The aforementioned factors were further exacerbated by the inflexibility in budgeting and human resource management, as well as the lack of autonomy at the division level in providing services and human resources required at different stages of the pandemic (i.e., protective equipment and an increased numbers of health providers). These results contribute to our understanding of the key organizational barriers to providing primary health care [15].

Oftentimes, strained and insufficient human resources aggravated the challenges involved in service delivery during the pandemic. At least during emergencies, flexible mechanisms for local budgeting and human resource management for PHC must be available in order to act according to changing contexts and needs [30]. Moreover, the pandemic highlighted the fragility of the health services’ human resources [13,31]. There are not enough health providers and, during the emergency, they faced multiple problems, such as few incentives, the quality of the workplace environment, and discrimination by the general population. CHWs were key for communication outreach, and investment in them would produce benefits by improving their capacities in terms of technology and attention to emergencies. There is a fundamental need to invest in building up capacity, as this would improve service delivery in general and during emergencies [32].

Demand-side limitations were mostly due to fears of infection and misinformation about service availability. Moreover, external factors, such as transportation restrictions and even climatic events, impeded service users from going to their clinics. Authorities stressed how the “fear of contagion” deterred the use of the clinics for assistance. The triangulation of the survey data and interviews with pregnant women confirmed these fears. On the one hand, two in five households restrained from seeking healthcare due to fears of infection [12], and pregnant women avoided ANC visits, especially in the third trimester [14]. On the other, pregnant women expressed that their risk assessments involved emotional distress but also socioeconomic hardships, such as a loss of income [10]. Future studies should focus on the users’ decisions regarding other types of EHNS (i.e., non-communicable disease services) in order to add important information to improve our understanding of their reasons for the interruption of the use of health services. In contrast to the available information, the divisional directors might be placing too much emphasis on misinformation. In a survey of 959 adolescent girls in rural districts with a mean age of 15 years old, these participants, at least, were aware of symptoms, contagion, and prevention measures, and they reported that their trusted sources of information were newspapers, radio, or TV (90%); family, friends, and relatives (61%); community awareness activities (35%); social media (32%); and calls/SMS (23%) [33]. While more data from the general population is needed, these results suggest that, even in the presence of widespread misinformation, campaigns by trusted sources such as media outlets may be more effective than is immediately apparent. Moreover, diffusion of information campaigns relying on influential figures, such as community leaders (i.e., imams), NGOs, and family members can yield positive results [34], even if authorities do not acknowledge them.

Little information emerged on surveillance and service monitoring at the local level during the pandemic, even though divisional directors were in a key position to report on these internal mechanisms. Some directors complained about the lack of rapid and updated information and that no formal strategies were identified to fill these gaps. Information and monitoring systems for disease and service delivery need to be improved at the local level [35]. Although there was limited information in this area, the little evidence available suggested that information and monitoring systems need to be better articulated. A key reminder to take away from the pandemic is the need for robust communication infrastructure and protocols in order to effectively gather and promptly disseminate information [36]. These systems must move towards integrated digital solutions with the capacity to aid decision making, even if they demand building up training and capacity.

Nonetheless, the narratives also revealed efforts to adapt in terms of communication, coordination, and service provision. Adaptation was mainly achieved using mobile phones, social media, and ZOOM. Previous studies already noted these adaptations [25]; however, directors emphasized that their role was much more relevant than such studies suggest. These types of technology were used for service delivery, outreach services, and communication between providers, showing that these are important resources for health promotion and service delivery. Communication was a recurrent challenge, but given the ubiquity and affordability of mobile phones and social media, three types of promising innovations might help to complement traditional health services, to facilitate communication between health providers, and to aid in regular planning and coordination: (1) the use of mobile phones to complement service delivery; (2) the use of social media for information dissemination; and (3) the actual implementation of telemedicine and mHealth. However, such adaptations also require confidentiality safeguards for providers and patients, a minimal infrastructure, and training to clarify how and when to use these innovations [37].

While divisional directors acknowledged the importance of national guidelines, their specific roles in service delivery, communication, and surveillance at the local level remain unclear. A relevant finding was that such guidelines were not adapted to specific populations and, in some cases, might have also discouraged the use of services (such as the testing protocols required for pregnant women). These problems were shown to have the potential to increase inequities in the pandemic response [38]. Similarly, it was unclear how the monitoring of diseases and service delivery was used to support further decision making at the upazila or division levels. The dissemination of key messages could profit from digital resources (i.e., social media and mobile phones), as well as the traditional outlets available locally, including community health workers. Guidelines can be effective communication tools [39], but they need to be inclusive and follow implementation science principles, and their use should be carefully explained [40].

It has been recognized that community health workers (CHW) have been relevant actors in addressing the COVID-19 pandemic, because they contribute to the protection of health workers and vulnerable populations and to strategies for reducing contagion, and they help to maintain healthcare services [41,42]. In Bangladesh, the previous experience in grassroots work of CHWs was fundamental to promoting COVID-19 public health measures and health service promotion. Communities trusted CHWs, who were able to work with community and religious leaders for the purpose of communication outreach. Despite the positive performance of CHWs, the directors considered that their effect may have been limited in the context of misinformation about COVID-19 rapidly spreading through social media. The threat of misinformation was associated with a reduced likelihood to comply with health guidance measures and with vaccine hesitancy [43]; thus, it warrants the implementation of active and rapid mechanisms for counterbalancing its deleterious influence during emergencies. Likewise, partnerships with non-governmental organizations were important for filling gaps in areas such as protective equipment and human resources, amongst others [44]. However, the need for rapid responses oftentimes meant that collaborations with valuable partners were not a priority. Better coordination during emergencies is the key for a better response.

The present study has several limitations. Due to its focus on the perspectives of divisional directors, it was unable to identify additional demand-side limitations by, for instance, including EHNS users. Moreover, restrictions associated with the COVID-19 pandemic precluded the research team from capturing the additional perspectives of other key actors with valuable knowledge of the supply-side limitations, such as frontline healthcare workers and CHWs, or even different actors who could expand on the influences of external factors, such as transport limitations. The focus on the divisional directors reduced our ability to triangulate evidence, and we had to rely on published material. In addition, because of the need to present rapid findings to stakeholders in Bangladesh, we were unable to collect follow-up evidence, and this entails a risk of recall bias of the divisional directors, especially concerning the early stages of the pandemic. By only using the collected data, based on a census of the divisional directors at the end of the first year of the pandemic, we consider that we saturated most of the material and struck a balance between providing timely findings to contribute to urgent policy changes and the production of knowledge for academic audiences [19]. Therefore, the results reported in this article are limited to the divisional directors’ experiences at one point in time, which, we believe, nevertheless offer a unique and—until now—missing perspective on how EHNS were provided in a critical period of the pandemic in Bangladesh.

## 5. Conclusions

The experiences of the division-level directors during the first months of the pandemic in Bangladesh offer detailed accounts of the disruptions in EHNS delivery and community outreach provision, as well as some of the factors that reduced care-seeking, such as the fear of contagion. The directors highlighted several themes as key to their pandemic responses, such as human resource strains and shortages, the late arrival of protective equipment, constant misinformation, the role of specific training and guidelines for building capacity, the use of technology to communicate during lockdowns, and financing restrictions during the first year of the pandemic. Even though their accounts are anchored to a specific perspective during a specific period, the experiences in Bangladesh resemble those faced by health systems in other countries during the pandemic. While the experience of Bangladesh during the pandemic confirms concerns regarding the performance of the health system [3], it also reveals how these obstacles were common and provides lessons for countries facing similar challenges.

The facilitators, barriers, and adaptations that the divisional directors described help to identify lessons for other low- and middle-income countries with strained health systems facing crises such as the COVID-19 pandemic. A better pandemic response would entail flexible investments in health infrastructure and equipment (i.e., telemedicine and protective gear), modern monitoring systems, and the constant hiring and training of human resources, including both frontline workers and CHWs. The wellbeing of health providers should be prioritized. Likewise, greater organizational autonomy at the divisional level could lead to faster decisions that adapt to local contexts and to immediate problems. Moreover, outreach efforts by CHWs and partnerships with well-established NGOs can offer a powerful strategy that profits from previously established trust in their crisis management and service provision capacities. While misinformation is a great concern, there are communication strategies that use trusted sources of information and establish social networks to disseminate key messages. National guidelines can be a useful tool for standardizing practice; however, they need to be developed through the lens of equity in order to adapt to specific populations and require flexibility due to unexpected circumstances. The use of communication technology is a key adaptation with the potential to remain in place when the pandemic recedes, but this would require confidentiality safeguards, infrastructure, and training in order to be sustainable. Acting on these lessons offers the promising possibility of fostering preparedness, building capacity, and creating more resilient health systems.

Looking ahead, health systems in most countries will have to identify service delivery gaps generated by the pandemic, particularly in EHNS, and plan strategies for catching up. These actions will have to consider the restrictions that the COVID-19 continues to impose (i.e., human resources, physical infrastructure, surveillance, and medical supplies). Better health systems will stem from the adaptations to the new phases of the pandemic and from the lessons learned from this incredibly challenging period.

## Figures and Tables

**Table 1 healthcare-10-01619-t001:** Characteristics of the key informants.

Variables	
***Age***, mean (sd)	55.8 (1.8)
***Gender***, % (n)	
Female	12.5(2)
Male	87.5 (14)
***Education*** (highest degree), % (n)	
Medical doctor	50 (8)
Postgraduate	50 (8)
***Current position***, % (n)	
Divisional Director—Health	50 (8)
Divisional Director—Family Planning	50 (8)
***Years in the current position***, % (n)	
Less than 1 year	37.5 (6)
1–3 years	43.75 (7)
More than 10 years	6.25 (1)
Currently retired	12.5 (2)
***Duration of the interview***, mean (sd)	41.9 (16.2)

**Table 2 healthcare-10-01619-t002:** Barriers identified for service delivery by division.

Barriers for Service Delivery	Rangpur	Chittagong	Khulna	Barisal	Dhaka	Mymnesingh	Rajshahi	Sylhet
Fear of going to clinic or hospital								
Transportation restrictions								
Service disruptions and prioritization of COVID-19-specific care								
Health worker shortage								
Late arrival and low quality of protective equipment								
Scarce guidance on telemedicine								
Private clinics closed								
Resource restrictions								
Insufficient infrastructure								
Low connectivity								
Floods								
Stigmatization								
**Barriers for communication and community outreach**	Rangpur	Chittagong	Khulna	Barisal	Dhaka	Mymnesingh	Rajshahi	Sylhet
Misinformation								
Guidelines with insufficient information								
Guidelines not tailored to needs								
Outreach service interruptions								
**Barriers for surveillance and service monitoring**	Rangpur	Chittagong	Khulna	Barisal	Dhaka	Mymnesingh	Rajshahi	Sylhet
In-person supervision decreased								
Lack of information systems								
Outdated health surveys								
**Legend:**								
Demand Factors	6–8 Divisions			4–5 Divisions			2–3 Divisions	
Supply Factors								

## Data Availability

The data underlying this article were provided by the World Bank Group under license/by permission. Data will be shared on request by the corresponding author with the permission of the World Bank Group.

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
