# Peer review of "A Qualitative Assessment of the Essential Health and Nutrition Service Delivery in the Context of COVID-19 in Bangladesh: The Perspective of Divisional Directors"

_healthcare, 2022, doi:10.3390/healthcare10091619_

Round 1
Reviewer 1 Report (Previous Reviewer 1)
The manuscript has improved.
Reviewer 2 Report (Previous Reviewer 2)
I agree with the changes made by the authors and agree to their publication.
This manuscript is a resubmission of an earlier submission. The following is a list of the peer review reports and author responses from that submission.
Round 1
Reviewer 1 Report
The article is interesting, and it contains important information, but it is based on very low sample size, and I am not convinced with the data and reports. Authors must extend the sample size or explain the limitations in the discussion and what misinterpretations can be generated out of it. The discussion lacks reference citations, and it must be rewritten. The results section should be improved with more data or analysis.
Author Response
Dear reviewer,
Please find attached a document with the point-by-point response to each of your valuable comments.

Reviewer 2 Report
Recommendations
Methodology
In the methodology section the paragraph: Facilitates understanding of health outcomes, and their interactions with the health care delivery system, and seeks to improve clinical care and health systems (16). What does it refer to, the objective of the work or the qualitative research methodology? It is not clear
It is important to incorporate the informed consent of the interviewees and the approval of the Ethics Committee of the corresponding institution.
Results
The acronyms ANC and PNC, CHW refer to what, please specify in brackets.
Disclaimer
The bibliography used is very deficient and does not allow for discussion of the data with other studies, in fact there is not a single citation in the discussion.
Conclusions
The conclusion could be improved as it does not give a clear answer to the objective of the study, it should better specify the conclusion reached with this study.
Author Response

(The authors gave the same response as above.)
